# Simultaneous Voltammetric Detection of Acetaminophen and Caffeine Base on Cassava Starch—Fe₃O₄ Nanoparticles Modified Glassy Carbon Electrode

**Ani Mulyasuryani ***[ID]**, Rachmat Triandi Tjahjanto and Robi'atul Andawiyah**

Department of Chemistry, University of Brawijaya, Jl. Veteran 01, Malang 65145, Indonesia; rachmat_t@ub.ac.id (R.T.T.); obikandawiyah@student.ub.ac.id (R.A.)

**\*** Correspondence: mulyasuryani@ub.ac.id

**Abstract:** The new molecularly imprinted polymer (MIP) membrane based on cassava starch—Fe₃O₄— was developed to detect acetaminophen and caffeine simultaneously with the differential pulse voltammetry (DPV) method. Cassava starch was reacted with sodium tripolyphosphate (STPP) as a crosslinking agent, while acetaminophen and caffeine were added as templates. The Fe₃O₄ nanoparticles in the composite were added to increase the sensor's sensitivity. The experimental results show that the ratio between cassava starch:STPP:acetaminophen/caffeine in the mixture for MIP membranes influences the sensitivity of the sensor obtained. MIP membranes with the best sensitivity is produced at a mixture ratio of 2:2:1. The sensor performance is also affected by the pH of the solution and the type of buffer solution used. The sensor works very well at pH 2 in PB solution. Sensors produced from GCE modified with MIP membrane from cassava starch—Fe₃O₄ with acetaminophen and caffeine as templates have linear range concentrations, respectively, at 50–2000 μM and 50–900 μM. Sensor sensitivity was 0.5306 A/M against acetaminophen and 0.4314 A/M against caffeine with Limit of Detection (LoD), respectively, 16 and 23 μM. Sensor selectivity and sensitivity are better than those without MIP and can be applied for the determination of the content of acetaminophen in headache medicine, with an accuracy of 96–99% and with Relative Standard Deviation (RSD) 0.9–2.56%.

**Keywords:** Cassava starch; glassy carbon electrode (GCE); sodium-tripolyphosphate (STPP); electrochemical sensors; acetaminophen; caffeine

## 1. Introduction

In an effort to increase the selectivity and sensitivity of electrochemical sensors, sensors have been developed based on molecularly imprinted polymers and nanoparticles. Two natural polymers have been used as functional molecules in the development of molecularly imprinted polymer (MIP)-based sensors, namely chitosan and starch [1–3]. Cassava starch, which is a natural polymer, has the potential to be a functional polymer in the development of MIP-based sensors. Polymer molecules in cassava starch can be crosslinked by the addition of sodium tripolyphosphate (STPP) in a basic NaOH solution [4–6]. Based on these results, cassava starch can be used as a basic polymer for membranes in the development of electrochemical sensors for the detection of acetaminophen and caffeine, simultaneously. Increasing the sensitivity of a sensor can be done by adding a conductivity material, such as Fe₃O₄ nanoparticles.

Fe₃O₄ nanoparticles were chosen as additives in membranes because they can increase the electrical conductivity of polymers [7]. Fe₃O₄ nanoparticles are known to increase the electrical conductivity of polyindole/polyvinyl alcohol (PIN/PVA) composites. The electrical conductivity of PIN/PVA composites

is 1.25 mS/cm, while the electrical conductivity of PIN/PVA composites containing $Fe_3O_4$ nanoparticles is 100 S/cm [8]. Adding 0.1% $Fe_3O_4$ nanoparticles in the nata de coco membrane for potentiometric detection of phenol compounds can increase sensitivity up to 30% [9]. $Fe_3O_4$ nanoparticles have been used for the development of diazinon sensors using nata de coco membranes [10] and polyvinyl alcohol-based MIP membranes for the detection of chlorpyrifos [11] and monosodium glutamate [12].

Acetaminophen is one of the antipyretic and analgesic compounds that is most widely used as a safe and effective painkiller. Caffeine is an alkaloid which is a derivative of N-methyl from xanthine and is widely used to treat asthma, nasal congestion, and headaches. Headache medicines on the market generally contain a combination of acetaminophen and caffeine in amounts of 500–600 mg and 35–65 mg per tablet, respectively. Ensuring the quality of medicines circulating in the market, especially over-the-counter medicines, such as headache medications, is urgent. This is because there are several cases of the circulation of counterfeit medicines and repackaging expired medicines. The most important activity is determining the amount of active substances in these medicines. Determination of caffeine and paracetamol content in multicomponent medicines can be done by the titrimetric and the HPLC methods. The titrimetric method requires a long analysis time and is less sensitive, whereas the HPLC method has a high analytical sensitivity but it is considerably expensive [13–15]. Another method of analysis, such as spectrophotometry, without separation is less accurate because there are other substances in the medicines which might interfere the measurement. Therefore, this study seeks to develop a simultaneous, easier, quicker, and more accurate determination of acetaminophen and caffeine in headache medicine by using selective membranes based on cassava starch.

Acetaminophen is 4-N-acetamino phenol or commonly called paracetamol. It can be oxidized in an acidic environment [16] following the reaction as shown in Figure 1. Several electrochemical detection methods for acetaminophen have been developed, including the use of electrodes from Screen Printed Graphene Electrodes. Acetaminophen in biological samples can be determined using this sensor in the range of 0.1–50 μM at pH 7 [17].

**Figure 1.** Oxidation of acetaminophen in an acidic environment.

An acetaminophen detection method in medicines has also been developed using Multi Walled Carbon Nanotubes (MWCNT) nano material and chitosan–Cu complex compounds as sensors/working electrodes. These sensors can detect acetaminophen in the concentration range of 0.1–200 μmol/L at pH 7. To increase its sensitivity, ascorbic acid and dopamine are added [18]. The use of nano material with $CuO–CuFe_2O_4$ for the detection of acetaminophen and codeine, simultaneously, has resulted in a system with a high sensitivity where acetaminophen can be detected in the concentration range of 0.01–1.5 μmol/L. The detection was applied to biological fluid samples [19]. The ability to detect compounds, with performance at very low concentrations is not applicable for pharmaceutical samples. Detection of acetaminophen is best obtained by the voltammetry method both individually and simultaneously with the detection of other compounds [20–24].

Caffeine can be oxidized in an acidic environment by the mechanism shown in Figure 2. A study to develop a detection method has been carried out for electrochemical detection of caffeine using carbon paste as a working electrode and applied to pharmaceutical samples. In that experiment caffeine was detected in the concentration range of 1–80 μM [25]. The same detection method for caffeine has also been developed but it uses carbon electrodes, in the concentration range of 20–100 μM, and is

applied to beverage samples [26]. The development of electrochemical sensors for the detection of caffeine, both individually and simultaneously with other compounds, has been widely developed but is generally applied in beverage samples [27–31].

$$+H_2O \quad -2H^+ \ -2e \qquad + H_2O \quad -2H^+ \ -2e$$

**Figure 2.** Steps of the caffeine oxidation reaction in an acidic environment, proposed by Tadesse, Y., et al [25].

Several electrochemical sensors have been developed based on MIP to detect caffeine and acetaminophen individually [32,33]. Utilizing natural polymers from renewable materials as functional molecules in the preparation of MIP is an effort to produce low-cost and environmentally friendly sensors. Research on the development of MIP-based electrochemical sensors from cassava starch—$Fe_3O_4$—is a new breakthrough to produce sensors that are inexpensive, selective and sensitive and environmentally friendly.

## 2. Materials and Method

### 2.1. Materials

Chemicals used in the experiments were acetaminophen standard (Sigma Aldrich, Surabaya, Indonesia), caffeine standard (Sigma Aldrich), Britton Robinson (BR) buffer solution (pH 2–7), phosphate buffer (PB) solution (pH 7), cassava starch (local product), sodium tripolyphosphate for synthesis, sodium tripolyposphate STPP (Sigma Aldrich), glycerol for synthesis (Sigma Aldrich), sodium hydroxide (Merck), ethanol (Merck), $Fe_3O_4$ 50–100 nm (Sigma Aldrich).

The tools used in this study were glassy carbon working electrodes (GCE) with a disk diameter of 5 mm (Metrohm RDE.GC50) with electrode shafts, galvanostat potentiostat (Autolab PGSTAT204), Shimadzu 8400S fourier transform infrared spectroscopy (FTIR) scanning electron microscopy (SEM) FEI Inspect S50, pH meter Senz TI-13MO597, Yenaco YNC-OV-30L oven, shakers and glassware.

### 2.2. Preparation of MIP Membrane

The procedure for making sensors was adapted from experiments conducted by da Silva, N., Sechi, M., Teixeira, P.M. (2017) [6]. We added 2 g of cassava starch to boiling water, and stirred until the volume reached 100 mL (2% w/v), then a few drops of 0.1 M NaOH solution were added to the pH value of 10. Next, 22 mL STPP was added along with 11 mL acetaminophen and 11 mL caffeine (concentrations of acetaminophen and caffeine as in Table 1). The mixture was stirred at 80 °C for eight hours, then placed in a petri dish and dried at 80 °C overnight. The formed membrane was removed and washed with 3 × 30 mL of ethanol until it is free of acetaminophen and caffeine, confirmed spectrophotometry. The length of time for each washing step was one hour while shaken at 200 rpm. The washed membrane is then dried in the oven for two hours at 50 °C. The membranes were made in three different compositions, shown in Table 1.

**Table 1.** Compositions of membrane for electrode modification.

| Membranes | Percentage/(w/v) | | | |
|---|---|---|---|---|
| | **Cassava Starch** | **STPP** | **Acetaminophen** | **Caffeine** |
| M211 | 2 | 1 | 1 | 1 |
| M221 | 2 | 2 | 1 | 1 |
| M222 | 2 | 2 | 2 | 2 |

### 2.3. Electrode Modifications

A 0.05 g of dry membrane, 2 mL of hot water (70 °C), and 15 μL glycerol was mixed and continuously stirred for 30 min to form a homogeneous solution (viscous solution). Meanwhile, the glassy carbon electrode (GCE) surface was rinsed with ethanol and dried at 50 °C for 30 min. Next, the GCE surface was sprayed with a membrane suspension, then dried at 50 °C for 1 hour 30 min. An electrode modification was also made by adding 11 μL suspension of $Fe_3O_4$ nanoparticles 0.1% (w/v).

### 2.4. Cyclic Voltammetry (CV) and Differential Pulse Voltammetry (DPV) Measurements

The measurement was carried out on acetaminophen (1 mM), caffeine (1 mM), and acetaminophen-caffeine solutions (1:1 mM), in a BR buffer of pH 7. The electrodes used as a reference in the measurements were Ag/AgCl (3 M KCl) and Pt wires used as auxiliary electrodes. The potential applied to the CV is −0.1–1.8 volts vs Ag/AgCl (3 M KCl), with a scan rate of 0.1 V/s. The potential applied to DVP is −0.3–1.6 volts with a scan rate of 0.01 V/s, amplitude modulation 0.025 V, modulation time 0.05 s, and interval time 0.5 s.

## 3. Results and Discussions

### 3.1. Characterization of the Modified Electrode

The first step in electrode modification is to determine the optimum membrane composition. The membrane composition is summarized in Table 1, where each membrane is then used to modify the surface of the Glassy Carbon Electrode (GCE), to obtain the electrodes GCE-M211, GCE-M221 and GCE-M222. Each of the three electrodes was evaluated based on a DPV voltammogram profile for each acetaminophen and caffeine solution of 1 mM in a BR buffer of pH 7. The results of the study are shown in Figure 3.

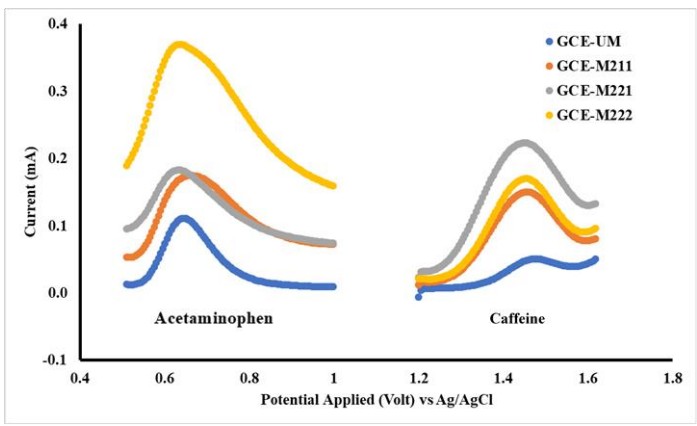

**Figure 3.** Differential pulse voltammetry (DPV) voltammograms for acetaminophen and caffeine 1 mM each in Britton Robinson (BR) buffer pH 7 measured by working electrodes: unmodified glassy carbon electrode (GCE-UM), GCE-M211 (GCE modified by M211), GCE-M221 (GCE modified by M221), GCE-M222 (GCE modified by M222).

As shown in Figure 3, the best voltammogram profile for acetaminophen is displayed by GCE-M211 and that for caffeine by GCE-M221. The peak current obtained by each electrode is shown in Figure 4, which shows that the highest peak current for acetaminophen is exhibited by GCE-M222, while for the caffeine is displayed by GCE-M221. Considering that caffeine levels in medicine sample samples are 10 times less than the level of acetaminophen, the GCE-M221 is chosen as the best electrode.

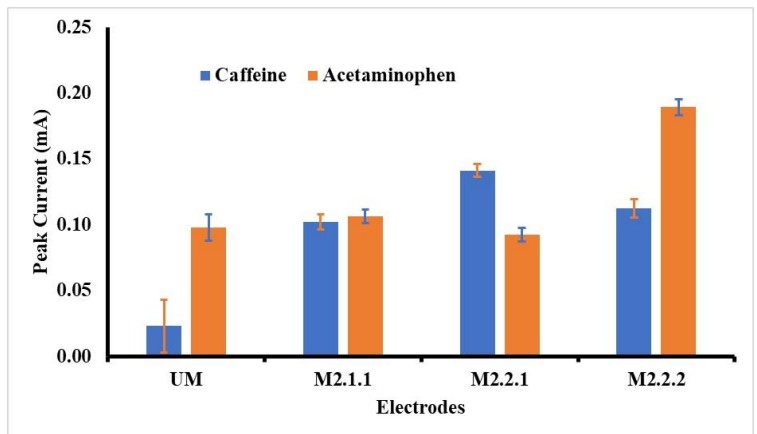

**Figure 4.** Observed peak currents for acetaminophen and caffeine 1 mM in BR buffer pH 7, on various working electrodes: GCE-UM (unmodified GCE), GCE-M211 (GCE modified by M211), GCE-M221 (GCE modified by M221), GCE-M222 (GCE modified by M222).

The peak potential (Table 2) for acetaminophen shifts slightly negative but remain in a tolerable range. As for caffeine, there is no difference in peak potential for all modified GCEs, although there is a shift towards negative compared to GCE-UM. This occurred allegedly because GCE-UM is less sensitive to caffeine. By observing the data summarized in Table 2, it can be concluded that there is no chemical interaction between acetaminophen and caffeine with the modified cassava starch membrane. Structural changes that occur in cassava starch modified with STPP can be studied from the FTIR spectrum (Figure 5). In the FTIR spectrum of cassava starch -modified, new peaks appear at wave numbers of 1556 and 1512 $cm^{-1}$, possibly correspond to P–O or P=O strain, and at 895 $cm^{-1}$ that belongs to the P–H strain.

**Table 2.** Peak potential for oxidation of acetaminophen and caffeine using working electrodes: unmodified glassy carbon electrode (GCE-UM), GCE-M211, GCE-M221 and GCE-M222.

| Electrode | Peak Potential/(volt) vs. Ag/AgCl | |
|---|---|---|
| | Acetaminophen | Caffeine |
| GCE-UM | 0.646057 | 1.47186 |
| GCE-M211 | 0.661163 | 1.45676 |
| GCE-M221 | 0.630951 | 1.45676 |
| GCE-M222 | 0.630951 | 1.45676 |

The $Fe_3O_4$ nanoparticle was added into the modified cassava starch membrane to increase sensitivity. The $Fe_3O_4$ can interact with the hydroxyl groups in the modified starch molecule so that the membrane can form three-dimensional structures with wider pores. The presence of $Fe_3O_4$ can be identified by SEM photographs (Figure 6), in which the membrane surface containing $Fe_3O_4$ appears brighter than the background image. Figure 6a shows the membrane surface which is not added with the $Fe_3O_4$ nanoparticles. The CV and DVP voltammograms for a mixture of acetaminophen and caffeine using GCE-M221 and GCE-M221-$Fe_3O_4$ electrodes are shown in Figure 7. The increased peak current in the voltammogram suggests that $Fe_3O_4$ can increase sensitivity, both for acetaminophen and caffeine.

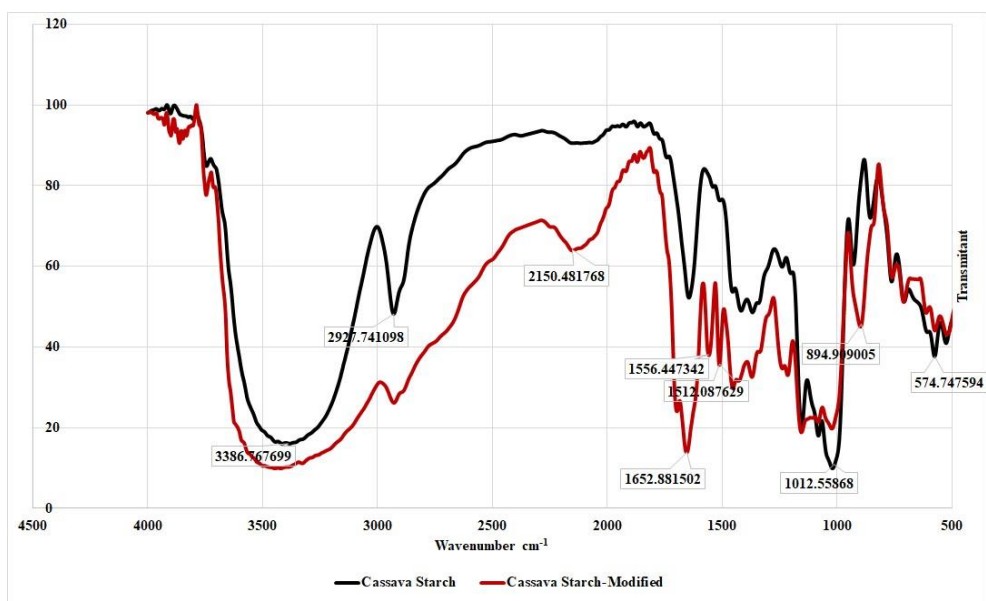

**Figure 5.** Fouried transform infrared spectroscopy (FTIR) spectrum of cassava starch and cassava starch modified by sodium tripolyphosphate (STPP).

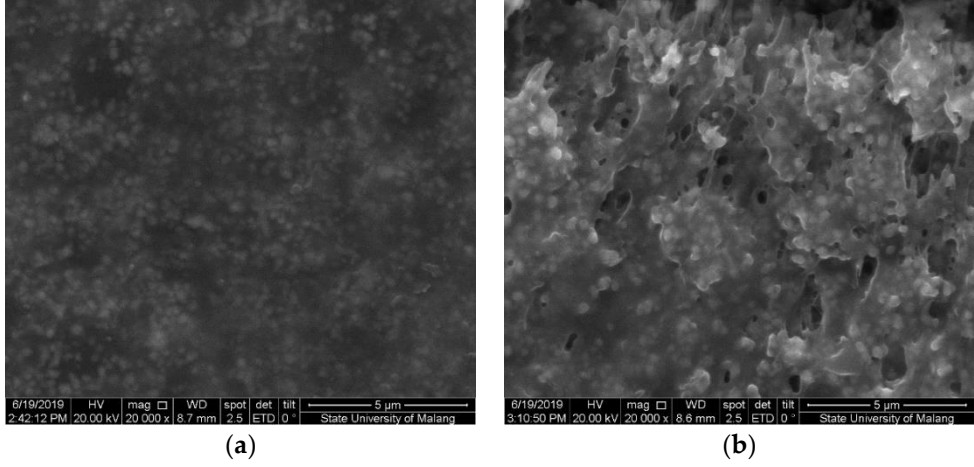

**Figure 6.** Scanning electron microscopy (SEM) image of membranes M221 (**a**) and M221-Fe$_3$O$_4$ nanoparticles (**b**).

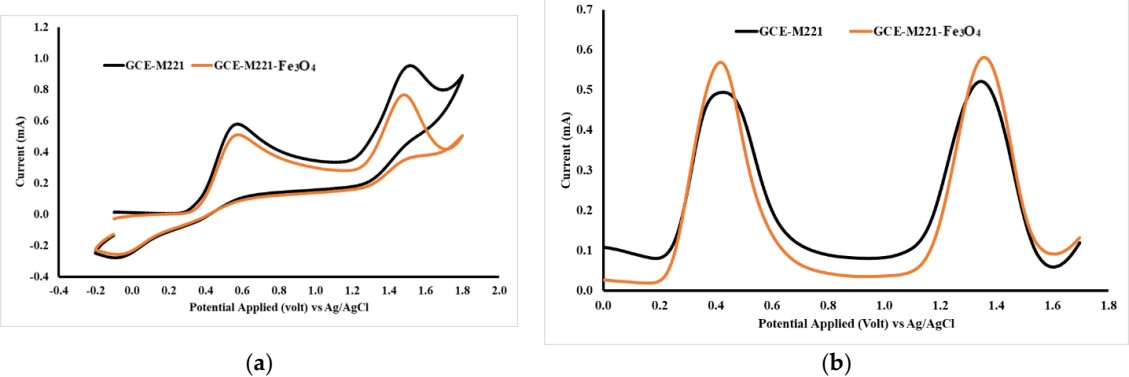

**Figure 7.** Cyclic voltammetry (CV) (**a**) and DPV (**b**) acetaminophen and caffeine voltammograms, 1 mM each in BR with a pH of 7, measured by working electrodes of GCE-M221 (GCE modified by M221), GCE-M221-Fe$_3$O$_4$ (GCE modified by M221 and Fe$_3$O$_4$ nanoparticles).

The oxidation of acetaminophen and caffeine is influenced by the pH of the solution. The experimental results shown in Figure 8 and summarized in Table 3 show that increasing the pH value can shift the oxidation potential to a more negative direction, both for acetaminophen and caffeine. Differential pulse voltammetry (DPV) data shows that the highest peak current occurs at pH 2, thus subsequent DPV measurements are carried out at a pH 2. The condition of the electrolyte solution affects the oxidation potential and peak current; therefore, these experiments are carried out in two kinds of buffer solution pH 2, which are BR and PB. DPV data (Table 4) shows that in the PB buffer solution, the caffeine peak current is slightly higher than that of in BR solution and the peak potential was shifted in a positive direction.

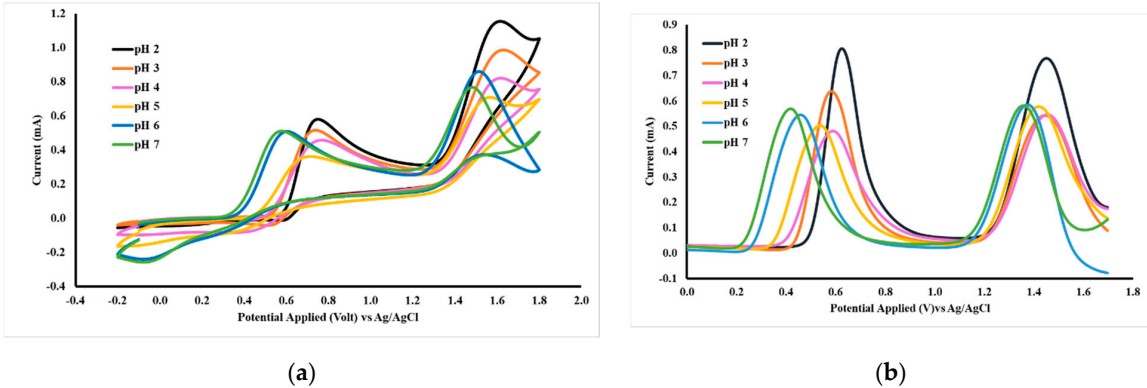

|       (a)       |       (b)       |

**Figure 8.** CV (**a**) and DPV (**b**) voltammograms of 1 mM acetaminophen and caffeine in BR pH 2–7, respectively, measured with a working electrode of GCE-M221-$Fe_3O_4$.

**Table 3.** Peak potential (Ep) and peak current (Ip) of acetaminophen and caffeine 1 mM in various Britton Robinson (BR) buffer solutions pH 2–7.

| pH | $E_P$/(Volt) vs. Ag/AgCl | | $I_p$/($\mu$A) | |
|----|---------------|----------|---------------|----------|
|    | Acetaminophen | Caffeine | Acetaminophen | Caffeine |
| 2  | 0.6215 | 1.4452 | 598 | 637 |
| 3  | 0.5711 | 1.4351 | 582 | 468 |
| 4  | 0.5216 | 1.4099 | 396 | 455 |
| 5  | 0.4712 | 1.3998 | 428 | 457 |
| 6  | 0.4166 | 1.3948 | 542 | 506 |
| 7  | 0.4079 | 1.3847 | 544 | 519 |

**Table 4.** Potential peak (Ep) and peak current (Ip) of acetaminophen and caffeine 1 mM in buffer pH 2, BR 0.04 M and PB 0.04 M.

| Compounds | $E_p$/(volt) | | $I_p$/($\mu$A) | |
|-----------|--------|--------|-----|-----|
|           | BR     | PB     | BR  | PB  |
| Acetaminophen | 0.6445 | 0.6647 | 444 | 425 |
| Caffeine      | 1.4746 | 1.4855 | 438 | 610 |

### 3.2. Quantitative Analysis

The quantitative relationship between the concentrations of acetaminophen and caffeine at peak currents is shown in Figure 9b, with a DPV voltammogram (Figure 9a). The concentrations of acetaminophen and caffeine are made equal in the range of 50–900 $\mu$M. The parameters of analysis for acetaminophen and caffeine by DPV using GCE-M221-$Fe_3O_4$ are shown in Table 5.

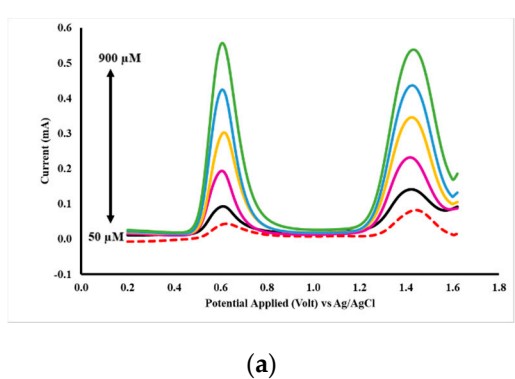
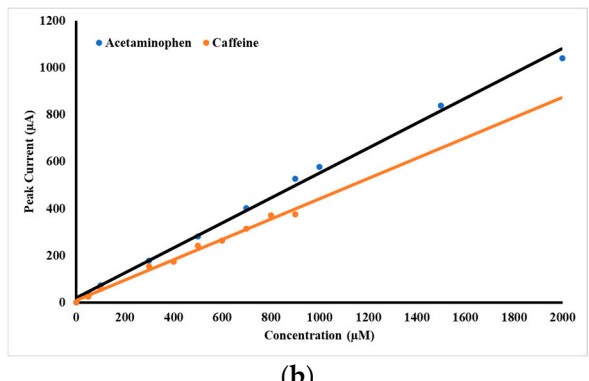

(**a**)           (**b**)

**Figure 9.** DPV acetaminophen and caffeine voltammograms at concentrations of 50–900 μM in PB pH 2 (**a**) and the curve of the relationship of acetaminophen and caffeine concentrations to peak currents (**b**), measured by a working electrode of GCE-M221-Fe$_3$O$_4$.

**Table 5.** Analysis parameters of differential pulse voltammetry (DPV) with acetaminophen and caffeine with GCE-M221-Fe$_3$O$_4$ as a working electrode.

| No | Parameters | Compound | |
|---|---|---|---|
| | | Acetaminophen | Caffeine |
| 1 | Linear range concentration/(μM) | 50–2000 | 50–900 |
| 2 | Sensitivity/(A/M) | 0.5306 | 0.4314 |
| 3 | Limit of Detection/(μM) | 16 | 23 |

Figure 10 indicates that the sample contains substances that interfere with the peaks in the applied potential area, as observed in the GCE voltammogram. Thus, it is proposed that the modified cassava starch membranes are selective membranes. The formation of acetaminophen and caffeine templates causes the molecules that reach the surface of GCE are only acetaminophen and caffeine, in which these molecules then oxidized to the appropriate potential. The polymer molecules present in the cassava starch reacted with STPP have been cross-linked (Figure 6) in a pattern that matches the molecular structure of acetaminophen and caffeine. When washing the membrane with ethanol, both acetaminophen and caffeine dissolve and leave traces of the appropriate shape of the associated molecules. Furthermore, Fe$_3$O$_4$ nanoparticles can increase sensitivity because they can interact with the -OH groups in the starch and form more rigid three-dimensional structures, (Figure 6b). Thus, it can increase the number of acetaminophen and caffeine molecules that reach the surface of the GCE.

The validity of the method is confirmed from the accuracy of the analysis results, the results of the analysis of acetaminophen and caffeine, simultaneously, with GCE-M221-Fe$_3$O$_4$ as a working electrode on DPV, is presented in Table 6. The accuracy of the analysis results appears from an error of less than 2%, except for acetaminophen in the "O" sample. The accuracy for acetaminophen in all samples was better compared to that for caffeine. This happens because the concentrations of caffeine are 6–9 times less than that of acetaminophen.

The application of the electrodes for measurements was carried out on headache medicines from three commercially available samples, namely samples P, B and O. Accuracy testing was done by comparing the levels of acetaminophen and caffeine analysis results with levels in the sample, based on what was stated on the label. Measurement of samples using unmodified GCE and GCE-M221-Fe$_3$O$_4$, and the obtained voltammograms are shown in Figure 10.

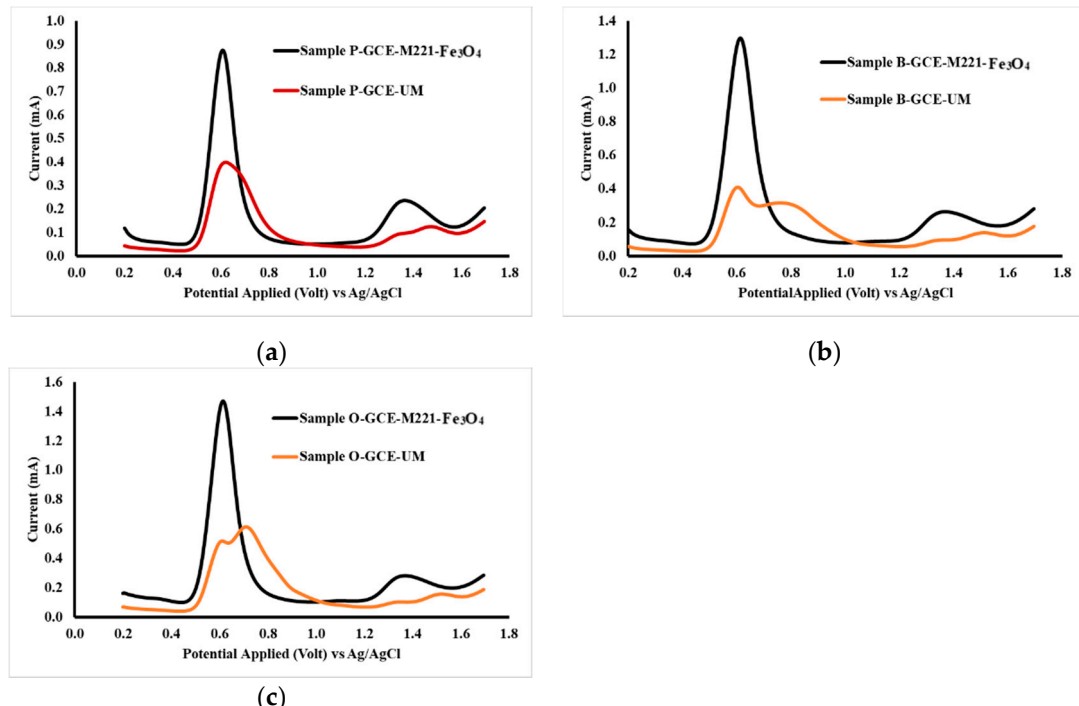

**Figure 10.** DPV voltammogram sample P (**a**), sample B (**b**) and sample O, (**c**) Each sample was dissolved in PB pH 2, measured both by GCE-M221-Fe$_3$O$_4$ and unmodified GCE as the working electrode.

**Table 6.** Recovery of acetaminophen and caffeine from P, B, and O samples dissolved in PB pH 2, determined by DPV with GCE-M221-Fe$_3$O$_4$ as the working electrode.

| Sample | Real mg/tablet | | Recovery mg/tablet | | Recovery (%) | |
|---|---|---|---|---|---|---|
| | Acet * | Caffeine | Acet * | Caffeine | Acet * | Caffeine |
| P | 600 | 65 | 591 | 64 | 98.50 ± 1.20 | 98.40 ± 2.02 |
| B | 600 | 50 | 587 | 48 | 97.83 ± 1.02 | 97.74 ± 1.95 |
| O | 500 | 35 | 480 | 34 | 96.11 ± 0.90 | 98.66 ± 2.56 |

* Acetaminophen.

## 4. Conclusions

Cassava starch can be used as a functional polymer in the development of electrochemical sensors for simultaneous detection of acetaminophen and caffeine, in headache medicine samples. The ratio of starch Manihot composition:STPP:acetaminophen/caffeine in the process of making MIP membrane affects the sensitivity of the sensor. The best MIP membrane is produced at a ratio of 2:2:1. Fe$_3$O$_4$ nanoparticles are proven to increase sensor sensitivity. The performance of the sensors is influenced by the pH and type of buffer solution. The sensor works optimally at pH 2 in PB solution. Sensors obtained from GCE modified with MIP cassava starch-Fe$_3$O$_4$ membrane have better selectivity and sensitivity and can be applied to headache medicine samples, with 96–99% accuracy and RSD 0.9–2.56%.

**Author Contributions:** A.M. designed and set up experiment, data analysis, and writing, R.T.T. data analysis and writing, R.A. experimental work.

**Funding:** The research no external funding.

**Acknowledgments:** The authors would like to thank to University of Brawijaya for funding research via associate professor research grant 2019.

**Conflicts of Interest:** The authors declare no conflict of interest.

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
