# Peer review of "Simultaneous Voltammetric Detection of Acetaminophen and Caffeine Base on Cassava Starch—Fe3O4 Nanoparticles Modified Glassy Carbon Electrode"

_chemosensors, doi:10.3390/chemosensors7040049_

Round 1

Reviewer 1 Report

The authors prepared glassy carbon electrodes modified with a biopolymer (starch), Fe2O3 and MIPs for voltammetric quantification caffeine and acetaminophen in pharmaceuticals. The improvement in sensitivity over the unmodified glassy carbon was clearly demonstrated so that the proposed electrode and method may find application in the quality control of the pharmaceutical industry. However, in the last paragraph of the introduction, the authors stated that there are no previous papers describing the use of MIP-based electrodes for the studied compounds. This statement is not correct. Just make a search in Web of Science using the keywords caffeine and voltammetry and MIP. The same applies to acetaminophen. Thus, the manuscript can be improved if the authors compare the performance of the proposed sensor with that other MIP-based sensor already described in the literature.

Specific points:

Lines 109 – 111: Boiling water maybe not at 100 °C. It depends on the altitude. This procedure is not clear. Do you add 2 g of cassava starch to 100 mL of boiling water and then NaOH to pH10?

Line 111 - 112: … 11 mL of caffeine. What are the concentrations of the STTP and caffeine solutions?

Line 131: …. the potential was scanned from -0.80 to 1.8V vs Ag/AgCl (3M KCl) ...

Line 132: …. the potential was scanned from -0.3 to 1.6V vs Ag/AgCl (3M KCl) ...

Figure 3: You do not need to give the potential ranges in the legend because we are seeing them in the graph

Figure 4 and Table 2: Here you are repeating information that is in Figure 3. There is redundancy in the information.

Line 166: This is not clear. I see the potential being displaced to more positive (GCE-M211) or less positive (the others) values; What do you mean with "toward reduction"?

Line 167: This is not clear as well. I see the potential being displaced to less positive values; Again, what do you mean with "toward reduction"?

Line 171: What is CS?

Line 178: change “….which is appear…” to “…… which appear ….”

Line 233: … namely samples P, B, and O.

Line 260: This is not a true validation. Validation involves a larger number of parameters such as limits of detection and quantification, selectivity, linearity, precision, robustness, besides to the accuracy, which in the present work, was evaluated by comparing the values found by the proposed method with those given in the label of the flask. This is a weak check of accuracy. Ideally, you should use another technique such as HPLC.  Alternatively, spiking/recovering tests should be made to prove the accuracy of the method.

Table 6: Are the samples liquid or solid? I do not believe units in the label are in mol. Thus, I suggest you make the comparisons using the same units as in the labels.

There is an exaggerated number of significant figures in these % recoveries and in their respective deviation.

Conclusion: Manihot starch and cassava starch are synonymous? Why the change in the terminology?

Author Response

Dear Sir

Thank you for taking the time to review our manuscript. We have attached our response

Reviewer 2 Report

The article describes electrochemical sensors for simultaneous detection of acetaminophen and caffeine in model systems (pH =2-7) and in headache medicines from three commercially available sources. Optimal membrane composition was found, Fe3O4 nano particles were added to increase sensitivity. Linear range of concentration, LOD and sensitivity for acetaminophen and caffeine were determined. 

The topic of the article is applicable. Minor spelling check is required.

Author Response

Dear Sir,

Thank you for taking the time to review our manuscript. We have attached our response

Reviewer 3 Report

The work is interesting, well structured and with relevant information. I think it will be of interest to readers. The current version requires minimal changes to be published

Author Response

(The authors gave the same response as above.)

Reviewer 4 Report

chemosensors-610402

Simultaneous Voltammetric Detection of 2 Acetaminophen and Caffeine Base on an Cassava Starch – Fe3O4 Nanoparticle Modified Glassy Carbon Electrode

Reviewer 1

The manuscript presents the simultaneous voltammetric detection of acetaminophen and caffeine base on a cassava starch – Fe3O4 nanoparticle modified glassy carbon electrode. Some aspects should be clarified before its publication.

Please check the English.

Materials and Method.

Electrode modifications

Page 4

The authors state that: a dry membrane was added to hot water and stirred to form a homogeneous solution.

Please specify how the authors checked that by spraying the membrane suspension result in PIM.

Results and discussions

Electrode modifications

Page 5

Fig. 4: Please specify from which experiments the peak currents are represented.

Page 7

Fig. 8. Please check the legend of the two figures, because there is not a concordance. It seems that in Fig. 8a the black curve should be for pH = 2.

Conclusion

Page 10

Please be consistent with the name of the starch throughout the manuscript.

Author Response

(The authors gave the same response as above.)

Round 2

Reviewer 1 Report

The manuscript was properly revised. I think it is suitable for publication.